# Conserved RNA-binding specificity of polycomb repressive complex 2 is achieved by dispersed amino acid patches in EZH2

Yicheng Long[1], Ben Bolanos[2†], Lihu Gong[3,4†], Wei Liu[2], Karen J Goodrich[1], Xin Yang[3,4], Siming Chen[3,4], Anne R Gooding[1], Karen A Maegley[5], Ketan S Gajiwala[2], Alexei Brooun[2*], Thomas R Cech[1*], Xin Liu[3,4*]

[1]Department of Chemistry and Biochemistry, BioFrontiers Institute, Howard Hughes Medical Institute, University of Colorado Boulder, Boulder, United States; [2]Worldwide Medicinal Chemistry, Worldwide Research and Development, Pfizer Inc., San Diego, United States; [3]Cecil H. and Ida Green Center for Reproductive Biology Sciences, Division of Basic Research, Department of Obstetrics and Gynecology, UT Southwestern Medical Center, Dallas, United States; [4]Department of Biophysics, UT Southwestern Medical Center, Dallas, United States; [5]Oncology Research Unit, Worldwide Research and Development, Pfizer Inc., San Diego, United States

*For correspondence:
Alexei.Brooun@pfizer.com (AB);
thomas.cech@Colorado.EDU
(TRC);
Xin.Liu@utsouthwestern.edu (XL)

†These authors contributed equally to this work

**Abstract** Polycomb repressive complex 2 (PRC2) is a key chromatin modifier responsible for methylation of lysine 27 in histone H3. PRC2 has been shown to interact with thousands of RNA species in vivo, but understanding the physiological function of RNA binding has been hampered by the lack of separation-of-function mutants. Here, we use comprehensive mutagenesis and hydrogen deuterium exchange mass spectrometry (HDX-MS) to identify critical residues for RNA interaction in PRC2 core complexes from *Homo sapiens* and *Chaetomium thermophilum*, for which crystal structures are known. Preferential binding of G-quadruplex RNA is conserved, surprisingly using different protein elements. Key RNA-binding residues are spread out along the surface of EZH2, with other subunits including EED also contributing, and missense mutations of some of these residues have been found in cancer patients. The unusual nature of this protein-RNA interaction provides a paradigm for other epigenetic modifiers that bind RNA without canonical RNA-binding motifs.
DOI: https://doi.org/10.7554/eLife.31558.001

## Introduction

Polycomb Repressive Complex 2 (PRC2) is essential for epigenetic silencing of gene expression during embryonic development and disease pathogenesis (*Schuettengruber and Cavalli, 2009*; *Margueron and Reinberg, 2011*; *Helin and Dhanak, 2013*). PRC2 specifically catalyzes mono-, di- and tri-methylation of lysine 27 on histone H3 (H3 K27), which is a hallmark for repressed chromatin and provides a chromatin structure that is repressive to transcription (*Cao et al., 2002*; *Czermin et al., 2002*; *Müller et al., 2002*; *Yuan et al., 2012*). The PRC2 core complex contains four subunits: (1) EZH2 (enhancer of zeste homolog 2) or its closely related homolog EZH1, (2) SUZ12 (suppressor of zeste 12), (3) EED (embryonic ectoderm development) and (4) RBBP4 (retinoblastoma binding protein 4) (*Margueron and Reinberg, 2011*). The complex can include additional

subcomponents such as AEBP2 (adipocyte enhancer-binding protein 2). EZH2 contains the enzymatic SET (Su[var]3–9, enhancer of zeste, trithorax) domain, which catalyzes the transfer of methyl groups from S-adenosylmethionine (SAM) to H3 K27 (*Rea et al., 2000*; *Dillon et al., 2005*; *Joshi et al., 2008*).

How human PRC2 is recruited to specific sites and regulated on chromatin in mammals is an important unanswered question. Multiple factors including chromatin marks, DNA sequence elements and other DNA-binding proteins (e.g. JARID2) have been suggested to recruit PRC2 to its genomic targets (*Lee et al., 2006*; *Martin et al., 2006*; *Ku et al., 2008*; *Margueron et al., 2009*; *Peng et al., 2009*; *Schuettengruber and Cavalli, 2009*; *Shen et al., 2009*; *Sing et al., 2009*; *Li et al., 2010*; *Pasini et al., 2010*; *Woo et al., 2010*; *Xu et al., 2010*; *Cuddapah et al., 2012*; *Yuan et al., 2012*; *Riising et al., 2014*). Although long noncoding RNAs (lncRNA) have been suggested to act as guides for recruitment of PRC2 to chromatin in cis or in trans (*Fitzpatrick et al., 2002*; *Mak et al., 2002*; *Plath et al., 2003*; *Silva et al., 2003*; *Rinn et al., 2007*; *Pandey et al., 2008*; *Zhao et al., 2008*; *Kanhere et al., 2010*; *Heo and Sung, 2011*), recent studies have emphasized the ability of RNA to inhibit PRC2 methylation of histones (*Cifuentes-Rojas et al., 2014*; *Kaneko et al., 2014*; *Beltran et al., 2016*; *Wang et al., 2017a*). Numerous RNAs were identified to interact with PRC2 in vivo using RNA immunoprecipitation approaches (*Zhao et al., 2010*; *G Hendrickson et al., 2016*). PRC2-binding RNAs include mRNAs as well as lncRNAs such as Kcnqtlot1, ANRASSF1, and COLDAIR (*Rinn et al., 2007*; *Pandey et al., 2008*; *Redrup et al., 2009*; *Heo and Sung, 2011*; *Beckedorff et al., 2013*).

PRC2 binds RNA promiscuously in vitro and in vivo (*Davidovich et al., 2013*, *2015*), meaning that it binds many natural RNAs with affinities within the same order of magnitude. This promiscuous binding can be explained by PRC2 recognizing short tracts of G's and G-quadruplexes, small motifs which are ubiquitous in the transcriptome (*Wang et al., 2017b*).

What elements within PRC2 bind RNA has been unclear, especially since none of the core PRC2 subunits has an RNA recognition motif (RRM). An unstructured region (residues 342–368) of mouse EZH2 that has been proposed to bind RNA (*Kaneko et al., 2010*) is not phylogenetically conserved (*Jiao and Liu, 2015*), and more important elements for RNA interaction are now apparent (see below). Separation of the RNA-binding role of PRC2 from its other activities – formation of the catalytically competent complex, recognition of histone tails, methyltransferase activity and interactions with accessory proteins – would greatly help to identify any PRC2 recruitment events that are dependent on RNA binding and to characterize RNA-mediated epigenetic silencing or inhibition of silencing.

A 21 Å electron microscopy structure of the human PRC2 complex shed light on its molecular architecture and subunit organization (*Ciferri et al., 2012*), and recently three high-resolution crystal structures of the minimal catalytic core of PRC2 (EZH2, EED and Suz12-VEFS) were determined. These include a complex from the fungus *Chaetomium thermophilum* (*ct*PRC2) (*Jiao and Liu, 2015*), a chimeric *ac/hs*PRC2 containing EZH2 from the American chameleon (*Anolis carolinensis*) and the remaining subunits from *H. sapiens* (*Brooun et al., 2016*) and an all-human PRC2 complex (*Justin et al., 2016*). Surprisingly, no canonical RNA-binding motif can be identified in these structures.

In this study, we find that the specificity of PRC2 for G-rich RNA and G-quadruplex RNA, previously shown for *hs*PRC2 (*Wang et al., 2017b*), is shared with *ct*PRC2, suggesting that the PRC2-RNA specificity could be conserved over a vast evolutionary distance. Surprisingly, while EZH2 emerges as the critical RNA-binding subunit for both *ct*PRC2 and *hs*PRC2, different sets of residues are involved. The manner in which RNA-binding elements are dispersed across the surface of PRC2 is unanticipated, and may portend similarly complex binding surfaces in other histone modification complexes and DNA methyltransferases that bind RNA.

## Results

### The active fungal *ct*PRC2 complex exemplifies similar RNA-binding specificity as human PRC2

The PRC2-RNA interaction has been exclusively studied in mammals, and expanding such knowledge to additional organisms would help to understand the biological importance of this interaction

through evolution. We therefore characterized the RNA-binding specificity of a crystallized PRC2 complex (ctPRC2) from a thermophilic fungus (*Jiao and Liu, 2015*). We first measured the binding affinity (illustrated by $K_d^{app}$) of purified ctPRC2 with a variety of synthetic homopolymeric 40-mer RNAs. ctPRC2 had specificity for poly (G) RNA, while having the lowest affinity for poly (A) RNA (*Figure 1A*). Second, ctPRC2 preferred binding to RNA sequences capable of folding into G-quadruplexes ((GGAA)$_{10}$, (G4A4)4 and G$_{40}$) (*Figure 1B and C*). For example, (GGAA)$_{10}$ and (GA)$_{20}$ RNA

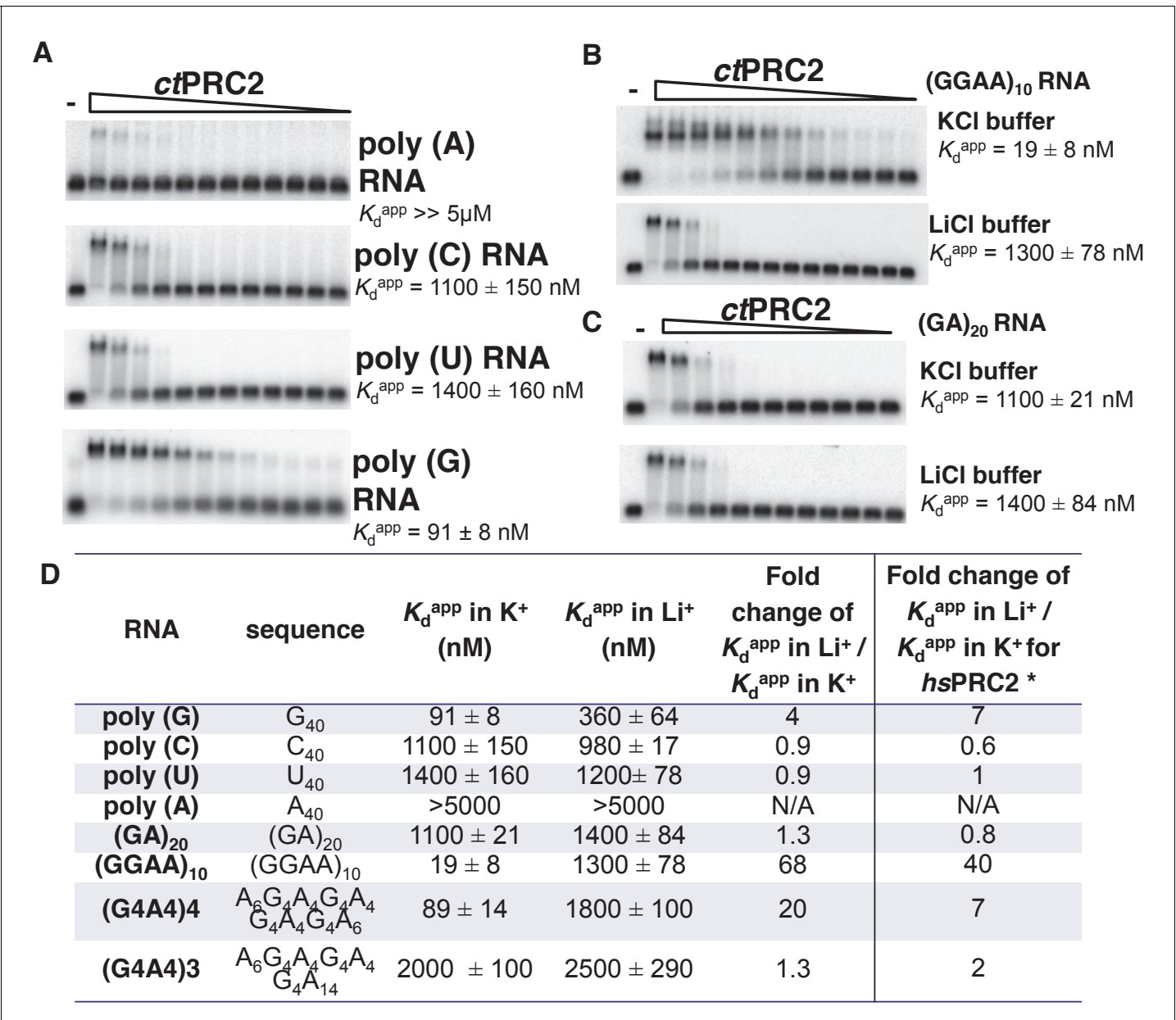

**Figure 1.** ctPRC2 prefers binding to G-rich RNAs and G-quadruplexes, and shares RNA-binding specificity with hsPRC2. (**A**) Binding of ctPRC2 to four 40-mer homopolymeric RNAs was tested using EMSA with standard 100 mM KCl-binding buffer. (**B**) Binding of ctPRC2 to a G-quadruplex-forming (GGAA)$_{10}$ RNA was tested in 100 mM KCl and 100 mM LiCl-binding buffers. (**C**) Binding of ctPRC2 to a control RNA (GA)$_{20}$ was tested in 100 mM KCl and 100 mM LiCl-binding buffers. (**D**) Comprehensive analysis of binding affinities of ctPRC2 to a variety of 40-nt RNAs. * Results of this column were from *Wang et al. (2017a)*. For (**A**), (**B**) and (**C**), ctPRC2 was used at successive threefold dilutions starting at 5 μM concentration or 4 μM for (GGAA)$_{10}$ RNA in KCl buffer. $K_d^{app}$ values and errors are mean and standard derivation of at least three binding experiments performed on different days and often by a different person.

DOI: https://doi.org/10.7554/eLife.31558.002

have the same nucleotide composition, but $ct$PRC2 has strong preference for $(GGAA)_{10}$, which is capable of forming G-quadruplexes. Third, the affinity for G-quadruplex-forming RNAs was cation-specific, as illustrated by the fold change of $K_d{}^{app}$ in $K^+$ buffer versus $Li^+$ buffer (**Figure 1D**). $K^+$ stimulates G-quadruplex conformation while $Li^+$ does not, so the fold change larger than one indicates a specific interaction between $ct$PRC2 and the folded G-quadruplex conformation of $(GGAA)_{10}$, $(G4A4)4$ and $G_{40}$ (**Figure 1D**). The cation specificity for binding to the G-quadruplex RNAs and lack of cation specificity for binding to other RNAs remarkably shows the same trend for $ct$PRC2 and $hs$PRC2 (**Figure 1D**).

## Identification of a minimal RNA binding complex for $ct$PRC2 (EBD-BAM of EZH2 + EED) that preserves specificity

Since $ct$PRC2 shares the binding specificity for G-quadruplex RNAs with $hs$PRC2, we first used $ct$PRC2 as a model to identify its RNA-binding residues using mutagenesis. Our initial approach involved site-directed mutagenesis of conserved positive-charged residues on the surface of $ct$PRC2, but 26 such mutations were found to have little or no effect on RNA-binding affinity (**Supplementary file 2**). We then conducted larger domain deletions and individual domain purifications to identify domains containing key RNA-binding residues. Based on the crystal structure, $ct$PRC2 consists of two halves - a regulatory moiety (EED and N-half of EZH2) and a catalytic moiety (VEFS domain of SUZ12 and C-half of EZH2) (**Jiao and Liu, 2015**). We expressed and purified each moiety separately and tested their binding affinities with the $(GGAA)_{10}$ RNA (**Figure 2**). Purified catalytic moiety tended to aggregate based on size-exclusion chromatography, but the regulatory moiety could be well purified using an *E. coli* expression system. Interestingly, the regulatory moiety had RNA-binding affinity comparable to the entire $ct$PRC2 complex, suggesting that this moiety could contain the major RNA-binding site (**Figure 2B**).

To further identify important domains for RNA binding, we expressed and purified a few well-defined domains from both the regulatory moiety and the catalytic moiety. For the components of the catalytic moiety, the VEFS and SANT2L domains showed little binding to $(GGAA)_{10}$ RNA ($K_d{}^{app} > 5$ µM), and the CXC-SET domains bound to $(GGAA)_{10}$ RNA with a 60-fold higher $K_d{}^{app}$ compared with wild-type $ct$PRC2. These results support the idea that the catalytic moiety may not contain the major RNA-binding sites. In contrast, in the regulatory moiety we identified a minimal RNA-binding complex consisting of EBD and BAM domains of EZH2 and the entire EED subunit, and it had RNA-binding affinity comparable to that of intact core $ct$PRC2. Strikingly, this minimal RNA-binding complex and the regulatory moiety maintained the binding specificity for G-quadruplex RNA (**Figure 2C**), tested by comparing $(GGAA)_{10}$ and $(GA)_{20}$ RNAs. This result indicated that this minimal RNA-binding complex contains the major RNA-binding site(s) that provides the specificity for G-quadruplex RNA. The minimal complex consists of EBD and BAM domains of EZH2 and the entire EED subunit, while EED alone binds RNA weakly ($K_d{}^{app} = 2.7$ µM). Addition of the EBD and BAM domains to EED increases the RNA binding by about 50-fold, suggesting that key determinants for RNA binding are located on these two domains of EZH2, with EED providing a platform for their proper 3D architecture.

## Four residues in BAM domain of $ct$PRC2 are important for RNA binding

The WD-repeat (WD: Trp-Asp dipeptide) protein EED provides a rigid scaffold for mutagenesis experiments on the minimal RNA-binding complex (**Figure 3A**). Thus, to investigate which region of EBD-BAM is critical for RNA binding, we tested various EBD-BAM deletions in the presence of EED to provide stability. We truncated the EBD-BAM peptide from either the N-terminus or the C-terminus in the context of the minimal RNA-binding complex (**Figure 3B and C**). When half of EBD was truncated from the N-terminus (N1 mutant), only a twofold reduction in RNA binding was observed. The second half of EBD has been shown before to be critical for EED interaction and further truncation causes dissociation of the complex (**Han et al., 2007**). In contrast, C-terminal truncation of the BAM domain (C1-5 mutants) led to a gradual reduction in RNA binding up to 17-fold when the BAM domain was completely deleted. These truncation experiments suggest an important role of the BAM domain in interaction with G-quadruplex RNA.

It is interesting that the BAM domain of EZH2 in $ct$PRC2 has a high electrostatic potential and contains a few positively charged residues that might be responsible for interacting with RNA. We

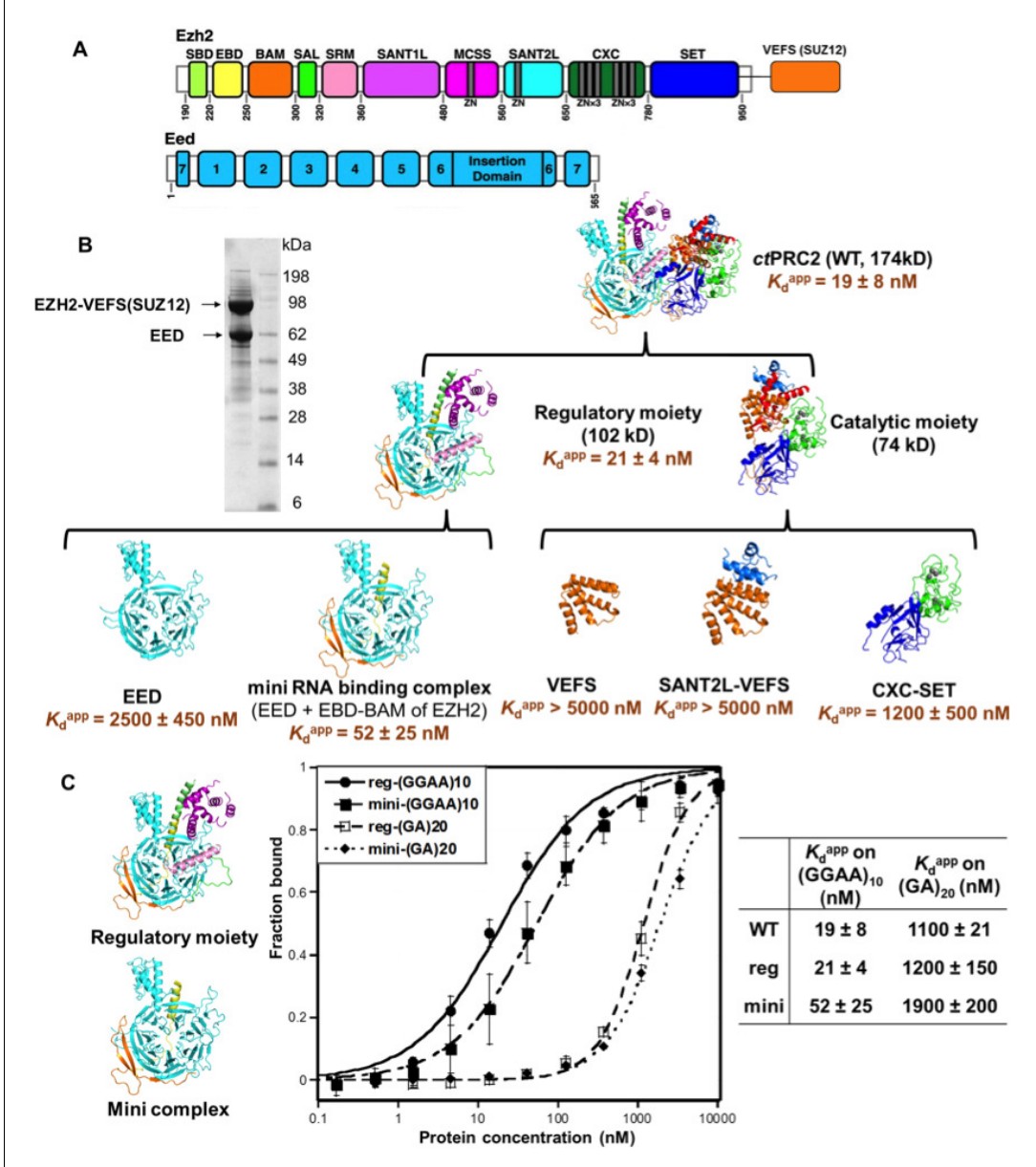

**Figure 2.** The minimal RNA-binding complex of *ct*PRC2 includes EED and EBD-BAM domains of EZH2, and additional affinity might be provided by the CXC-SET domains. (A) Purified *ct*PRC2 comprises two polypeptides: a 107 kDa peptide consisting of EZH2 fused with the VEFS domain of SUZ12, and a 67 kDa EED subunit. Individual domains are color coded as represented in structures of panel (B). (B) Purified *ct*PRC2 wild type was Coomassie stained on a SDS-PAGE gel (left). Sub-complexes of *ct*PRC2 were purified and tested for binding with $(GGAA)_{10}$ RNA (right). Regulatory moiety contains EED and N-terminal half of EZH2 (ending at SANT1L domain), and catalytic moiety contains C-terminal half of EZH2 (starting at MCSS domain) fused with VEFS domain of SUZ12. $K_d^{app}$ values and errors are mean and standard derivation of at least three binding experiments performed on different days. Minimal RNA-binding complex maintains the specificity of binding to G-quadruplex RNA. Regulatory moiety (reg) and minimal RNA-binding complex (mini; EED + EBD BAM of Ezh2) of *ct*PRC2. G-quadruplex-forming $(GGAA)_{10}$ RNA and control $(GA)_{20}$ RNA were tested for binding with the two complexes.

DOI: https://doi.org/10.7554/eLife.31558.003

mutated two patches of residues (mutant 1 and 2) in the context of the ternary *ct*PRC2 complex (EZH2-EED-VEFS of SUZ12) (*Figure 3D*); mutant 1 caused a three- to fourfold reduction in RNA-binding while mutant 2 had similar binding affinity as the wild type (*Figure 3E*). We further tested whether the integrity of the whole protein complex was affected by the mutations, through measuring their histone methyltransferase activities. Mutant 1 had similar methylation activity as the wild

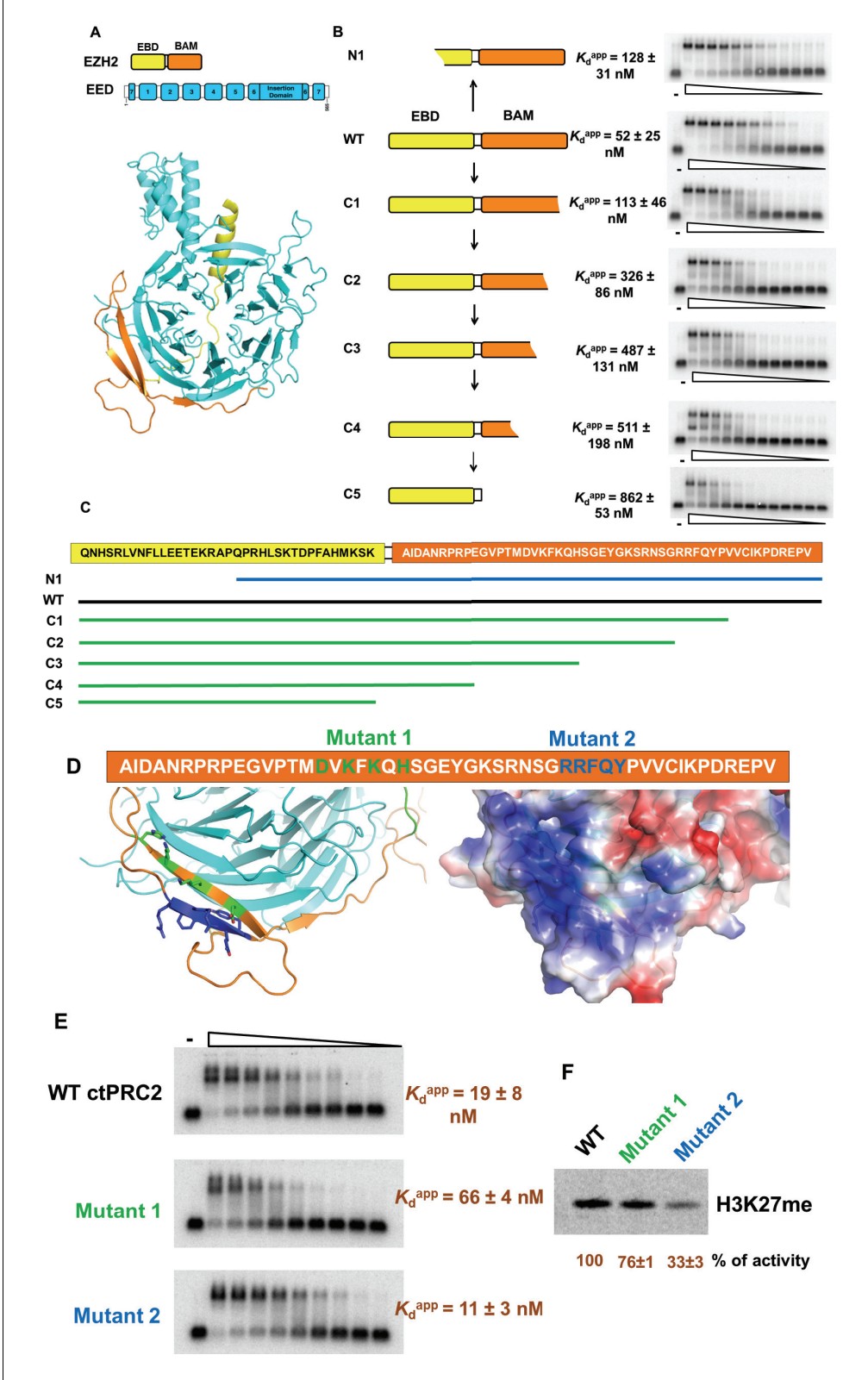

**Figure 3.** Four residues in BAM domain of *ct*PRC2 are important for RNA binding. (**A**) The minimal RNA-binding complex consists of EED and EBD-BAM domains of EZH2. (**B**) Mutagenesis study was conducted on the minimal RNA-binding complex, by trimming the EZH2 peptide from the N and C termini. Purified complexes were tested for binding to (GGAA)$_{10}$ RNA. Threefold titration of proteins was used in each EMSA experiment. Starting

*Figure 3 continued on next page*

*Figure 3 continued*

concentrations of each complex were: 10 µM for N1, C2, C3 and C4, 5 µM for wild type and C1, and 4 µM for C5. (C) Details of the EZH2 peptide sequence for wild type and the six mutants. Blue line indicates the N-terminal truncation mutant (**N1**), and green lines refer to the C-terminal truncation mutants (C1-5). (D) Two sets of residues (green and blue) in BAM domain were mutated in the context of the entire *ct*PRC2. Mutant 1 includes alanine substitution of four residues (shown in green), and mutant 2 includes mutation of five residues (shown in blue) to GGSGG sequence. These residues are also indicated in the structure and the electrostatic map. (E) Wild type and mutant *ct*PRC2 were tested for binding with (GGAA)$_{10}$ RNA, and each protein was titrated threefold from 500 nM. (F) Methyltransferase activity on histone H3 was tested for wild type and the two mutants. $^{14}$C-labeled SAM and purified H3 were incubated with each protein for 30 min and then resolved on a SDS-PAGE gel. Methyltransferase activity is indicated by the intensity of the radiolabeled H3 band. Activity is normalized to that of wild type, and the mean ±S.D. of three independent experiments is shown.

DOI: https://doi.org/10.7554/eLife.31558.004

type, while mutant 2 suffered greater reduction in catalytic activity (*Figure 3F*). Thus the better separation-of-function mutant of *ct*PRC2 is obtained by mutating four residues (D266, K268, K270 and H272) in its BAM domain.

## 3 m *hs*PRC2 (EZH2-EED-VEFS of SUZ12) contains the major RNA-binding regions in human PRC2

The amino acid sequence of the BAM domain is not well conserved between *ct*PRC2 and *hs*PRC2, and of the four key RNA-binding residues in the BAM domain of *ct*PRC2, only D266 is found in *hs*PRC2. Thus, the RNA-binding residues and sites on *hs*PRC2 could be different from *ct*PRC2, and we sought to identify the differences.

The only previously published RNA-binding site on a mammalian PRC2 subunit is residues 342–368 in an unstructured region of mouse EZH2 (*Kaneko et al., 2010*). However, when this sequence was deleted and tested in the context of an intact 4-mer PRC2 complex, binding affinity to a RepA RNA was not affected (*Figure 4—figure supplement 1*), and therefore other RNA-binding elements could exist outside of residues 342–368. Previously, we have shown that a human PRC2 5 m (5-subunit) complex (EZH2-EED-SUZ12-RBBP4-AEBP2) specifically recognizes G-quadruplex and other G-rich RNAs (*Wang et al., 2017b*), but it was unclear whether the human PRC2 3 m complex that was characterized structurally binds to RNA and shares similar specificity. We therefore purified the 3 m complex (full length human EZH2, EED (81-441) and SUZ12 VEFS S583D) using an insect cell expression system and tested its binding affinity on three different RNAs – (A)$_{40}$, (GA)$_{20}$ and (GGAA)$_{10}$, the last of which forms G-quadruplex structures. The 3 m *hs*PRC2 shared very similar RNA-binding specificity with the 5 m *hs*PRC2 complex previously characterized (*Figure 4A*), suggesting that the 3 m complex retains the major RNA-binding sites and could be a suitable model system for studies of PRC2-RNA interaction. Interestingly, the binding affinity for (GGAA)$_{10}$ RNA was fourfold lower for the 3 m relative to the 5 m complex, indicating that additional subunits in the 5 m may enhance the binding to G-quadruplex RNA or stabilize key RNA binding determinants compared to smaller PRC2 3 m complex.

## Human PRC2 deletion mutagenesis uncovers importance of N-terminal helix of EZH2 in RNA binding

We then pursued identification of key domains in the human 3 m complex for RNA interaction using a domain deletion mutagenesis approach similar to what we applied for *ct*PRC2. Besides the insect cell expression system used predominantly in this paper, in this section we expressed 3 m *hs*PRC2 wild type and mutants in *Saccharomyces cerevisiae* (all protein compositions and expression systems are shown in *supplementary file 1*); we verified that the (GGAA)$_{10}$ RNA-binding affinity of the 3 m *hs*PRC2 complex containing full length EZH2-VEFS of SUZ12 (543-695) fusion co-expressed with full length EED purified from yeast was comparable to that purified from insect cells (*Figure 4A and B*). A binary complex formed by SUZ12(ΔVEFS) and RBBP4 subunits bound (GGAA)$_{10}$ RNA very weakly (*Figure 4B* top right), further supporting that the major RNA-binding sites are located in the 3 m complex. When the previously published RNA-binding element (residues 342–368) in EZH2 (*Kaneko et al., 2010*) was deleted, binding to (GGAA)$_{10}$ RNA was not affected (*Figure 4B* middle

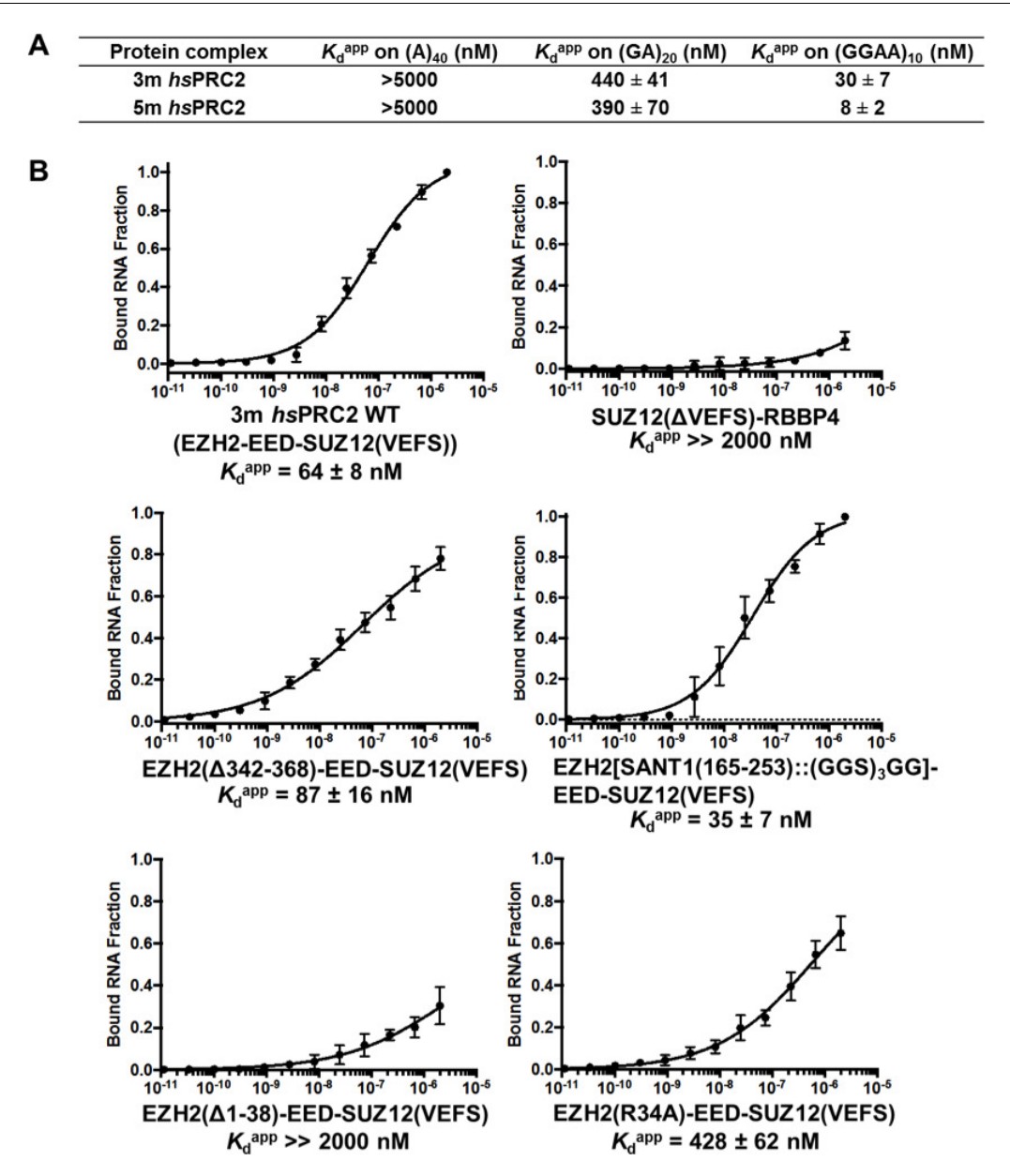

**Figure 4.** The crystallized 3 m *hs*PRC2 complex contains the major RNA-binding sites of *hs*PRC2, and the N-terminal helix of EZH2 plays an important role. (A) Previously tested *hs*PRC2 5 m complex and the crystallized 3 m core complex were expressed in insect cells, purified and tested for binding with (GGAA)$_{10}$, (GA)$_{20}$ and (A)$_{40}$ RNA. PRC2 proteins were titrated threefold starting at 5 μM concentration. (B) Mutagenesis analysis of the crystallized 3 m core complex uncovers the role of the N-terminal helix of EZH2 in RNA binding. Wild type and mutants were expressed using an *Saccharomyces cerevisiae* expression system. Wild type 3 m complex and the remaining part of the 4 m holoenzyme were tested for binding with (GGAA)$_{10}$ RNA (top panel). Three large deletion mutants and one point mutant (R34A) were tested for RNA binding (middle and bottom panels). $K_d^{app}$ values and errors are mean and standard derivation of at least three binding experiments performed at different days.

DOI: https://doi.org/10.7554/eLife.31558.005

The following figure supplements are available for figure 4:

**Figure supplement 1.** Deletion of residues 342–368 of EZH2 in mouse 4 m PRC2 does not affect binding to a RepA RNA.

DOI: https://doi.org/10.7554/eLife.31558.006

**Figure supplement 2.** SDS-PAGE gels and EMSA gels of *S.*

DOI: https://doi.org/10.7554/eLife.31558.007

left), supporting the mutagenesis results in the mouse 4 m PRC2 complex (4 m is EZH2-EED-SUZ12-RBBP4) (*Figure 4—figure supplement 1*). From the two published crystal structures of the 3 m *hs*PRC2 and *ac/hs*PRC2 complexes, we noticed that EZH2 contains a basic N-terminal helix adjacent to an acidic SANT1 domain. When the N-terminal helix was truncated by 38 residues, binding to $(GGAA)_{10}$ RNA was dramatically disrupted (*Figure 4B* bottom left, $K_d^{app} > 2$ μM), suggesting a potentially important role of this basic helix in RNA binding. However, the loss of RNA binding by this truncation could also be indirect and may be caused by charge repulsion after deletion of basic N-terminal helix and release of the adjacent *a*cidic SANT1 domain (surface electrostatic potential map was illustrated in *Brooun et al. (2016)*). When the adjacent SANT1 domain was replaced by a $(GGS)_3GG$ linker, RNA binding was slightly enhanced (*Figure 4B* middle right), plausibly by reducing the repulsive negative charge in this area or releasing the N-terminal helix to increase its flexibility to accommodate RNA binding. The R34 residue was found to be particularly important in the N-terminal helix (*Figure 4B* bottom right), because the R34A mutant exhibited a six- to sevenfold reduction in $(GGAA)_{10}$ RNA binding.

## HDX-MS identifies four regions in *hs*EZH2 that Are Perturbed by RNA Interaction

Deletion analysis and mutagenesis have the inherent risk of perturbing protein folding, so effects on RNA binding might be indirect as well as direct. As an alternative approach, we performed hydrogen deuterium exchange mass spectrometry (HDX-MS) analysis of the *hs*PRC2 3 m complex in the absence and presence of folded G-quadruplex RNA. $(GGAA)_{10}$ quadruplex was annealed as described previously, pre-incubated with *hs*PRC2 3 m complex at 8:1 molar ratio (RNA:3 m *hs*PRC2) prior to HDX studies. Deuterium uptake was measured at five time points over a period of 480 min for comparison of *hs*PRC2 against the RNA-*hs*PRC2 complex. A deuterium uptake profile of 258 (EZH2), 149 (EED) and 114 (SUZ12) peptides, generated from inline digest, provided 91.3%, 96.4%, 99.0% sequence coverage for each subunit, respectively.

We first examined the changes in deuterium uptake of the EZH2 subunit, which was previously shown to have highest baseline dynamics amongst the subunits of *hs*PRC2 3 m complex. Comparative deuterium uptake for the RNA bound and apo-EZH2 subunit highlighted four distinctly differentiated regions, 32–42 (Z1), 210–224 (Z2), 512–531 (Z3) and 645–665 (Z4) of human EZH2. Reduction in deuterium uptake for EZH2 in complex with $(GGAA)_{10}$ RNA suggests that these regions could be on the binding interface with G-quadruplex or impacted in an allosteric fashion (*Figure 5A*). Mapping these regions on the crystal structure of *ac/hs*PRC2 complex results in the following assignments: (1) Z1 is located in N-terminal alpha helix of EZH2 preceding EED binding site; (2) Z2 is located in the disordered region of SANT1; (3) Z3 is located in the disordered linker between SANT2 and cysteine-rich domain of *hs*EZH2 catalytic and also overlaps the beginning of the first Zn binding motif of EZH2 CXC; and (4) Z4 maps to i-SET region of EZH2 catalytic domain (*Figure 5B*).

Both EED (81-441) and SUZ12-VEFS S583D subunits of 3 m *hs*PRC2 exhibit lower baseline dynamics in the apo-state. While examining EED deuterium uptakes in the presence and absence of G-quadruplex RNA, we noticed two regions 290–308 (E1) and 338–350 (E2) with minor but statistically significant (p-value<0.05) changes in deuterium uptake (*Figure 5A*). Mapping of E2 (EED) region on the crystal structure of 3 m *ac/hs* PRC2 places it adjacent to Z1 site on the EZH2 subunit, suggesting that it could be part of putative G-quadruplex binding interface formed by EZH2-Z1, Z2 and EED-E2 regions (*Figure 5B*). Evidence of the inter-talk of these regions is that the N-terminal helix of EZH2 is bent at Z1 region in a stimulated state of *hs*PRC2 indicated in the crystal structure of a H3K27me3-stimulated 3 m *hs*PRC2 (*Justin et al., 2016*), bringing in Z2 region to approach the Z1, Z2 and E2 regions to make a more compact RNA binding site. Examining deuterium uptake of SUZ12-VEFS in the presence of G-quadruplex RNA revealed statistically significant uptake increase in the peptides spanning residues 617–629 (S1) (*Figure 5A*). SUZ12-VEFS S1 region is on the interface between SUZ12-VEFS and MCSS domain of EZH2 (*Figure 5B*). PRC2 regions with altered deuterium uptake profile could represent the direct G-quadruplex RNA binding interface, or reflect allosteric changes in PRC2 dynamics upon interaction with RNA.

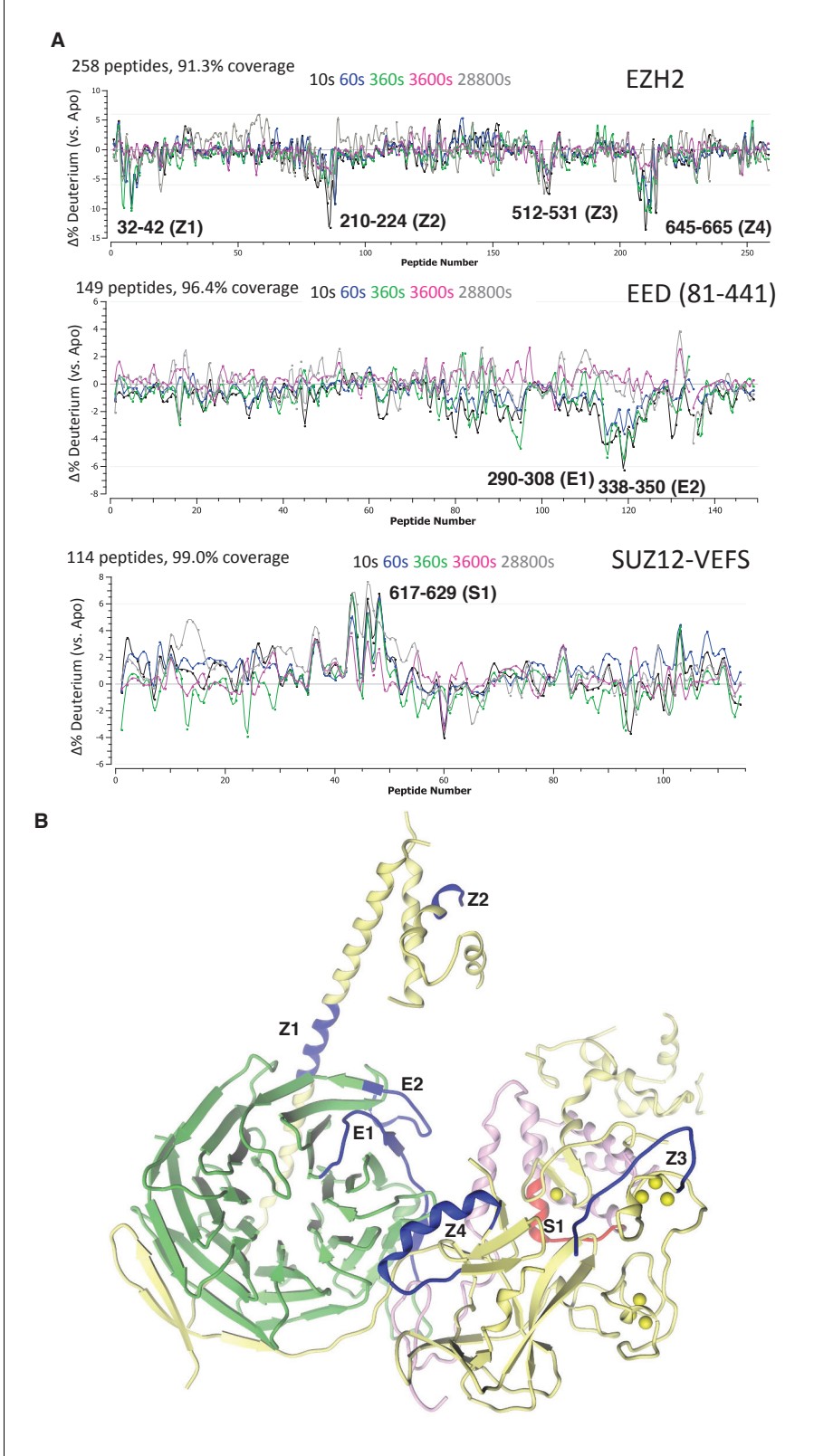

**Figure 5.** Regions protected by RNA binding identified using HDX-MS. (**A**) *hs*PRC2-(GGAA)$_{10}$ complex dynamics. Differential *hs*PRC2 vs *hs*PRC2-(GGAA)$_{10}$ (5 μM *hs*PRC2: 40 μM (GGAA)$_{10}$) in deuterium uptake plots of EZH2, EED (81-441) and Suz12-VEFS (545-726; S583D). Deuterium incubation times: 10 s, 1 min, 6 min, 1 hr, 8 hr. Peptides with statistically significant changes in deuterium uptake are labeled based on human protein numbering. (**B**) Mapping of regions with altered deuterium uptake in presence of (GGAA)$_{10}$ quadruplex onto the *ac/hs*PRC2 crystal structure.

*Figure 5 continued on next page*

*Figure 5 continued*

DOI: https://doi.org/10.7554/eLife.31558.008

The following figure supplement is available for figure 5:

**Figure supplement 1.** HDX-MS deuterium uptake heat map in the absence of RNA.

DOI: https://doi.org/10.7554/eLife.31558.009

## Identification of key residues on *hs*PRC2 for RNA binding

To verify the regions identified using the HDX-MS approach, we mutated the four regions of EZH2 and investigated the effects of these mutations on binding to $(GGAA)_{10}$ RNA. Alanine scanning mutagenesis was used to explore the role of specific residues in RNA binding in short structured Z1 and Z4 regions while NAAIRS (asparagine - alanine - alanine - isoleucine - arginine - serine [*Marsilio et al., 1991*]) sequence substitution was used for the longer and less structured Z2 and Z3 regions All mutants were expressed using the insect cell expression system and purified as described in Materials and methods section, and their methyltransferase activity was measured relative to the wild-type 3 m *hs*PRC2.

Alanine scanning mutagenesis of 32–42 (Z1) in *hs*EZH2 helped us to identify the most important residues in this region for RNA binding, including F32, R34 (also identified in the previous result section), D36 and K39. The K39A point mutation caused a 5-fold reduction in RNA binding, and the triple mutant [F32A D36A K39A] and the quadruple mutant [F32A R34A D36A K39A] led to additional reduction in RNA binding (*Figure 6A*).

For Z2 (210–224) of *hs*EZH2, we conducted NAAIRS sequence substitution mutagenesis to 208–213, and a moderate reduction in RNA binding was observed (*Figure 6B*). It is possible that Z2 might not directly contribute to RNA interaction and the change in Z2 deuterium uptake may be the consequence of an allosteric effect upon RNA binding.

Z3 (512–531) is adjacent to a positive-charged and unstructured region (489–510) that has no HDX-MS coverage (*Figure 5—figure supplement 1A*). RNA-binding studies with the chimeric 3 m *ac*/*hs*PRC2-3 protein complexes previously used for structural work (*Brooun et al., 2016*) showed that the substitution of residues in *ac*EZH2 corresponding to residues 488–510 in *hs*EZH2 to a glycine-serine linker caused a significant reduction in binding to $(GGAA)_{10}$ RNA, while deletion of the previously reported RNA-binding region (*Kaneko et al., 2010*) did not affect binding (*Figure 6—figure supplement 1B*). Additionally, an alternative RNA crosslinking approach using a 4-thio-uridine-labeled G-quadruplex RNA supports this region (residues 488–510 adjacent to Z3) as being critical for interaction, because mutation of this region in *ac*/*hs*PRC2 caused loss of crosslinking of the RNA to PRC2 (*Figure 6—figure supplement 1C*). This crosslinking experiment also indicated that the majority of RNA is crosslinked to the EZH2 subunit rather than the other two subunits, supporting our conclusion that EZH2 is the most important subunit for RNA interaction. Thus, we decided to focus NAAIRS scanning mutagenesis at the 489–510 region, which led to the identification of amino acids 489–494 (PRKKKR) as critical for RNA binding in the context of 3 m *hs*PRC2 complex. The 489–494 NAAIRS substitution mutant (PRKKKR to NAAIRS) retained its methyltransferase activity, with a 10-fold reduction in RNA binding. NAAIRS substitutions for other residues in the 489–510 region had only a modest effect on RNA binding and also had negative effect on methyltransferase activity suggesting that even if those binding effects are real they could be the consequence of a large conformation change.

Z4 (645–665) is located at the I-SET alpha-helix that is critical for maintaining catalytically competent PRC2 in basal state via interaction with SAL and this region interacts with SRM in the activated PRC2 structure resulting in further increase in catalytic activity. We picked two positive-charged amino acids (R653 and K656) in this region and substituted each of them with alanine. (We did not mutate R654 because it is involved in an intramolecular salt bridge and is important for catalysis.) The K656A mutant bound $(GGAA)_{10}$ RNA 4–5 fold weaker compared to the wild-type *hs*PRC2 concurrent with 50% increase of methyltransferase activity, while R653A had a modest change in RNA binding and also showed reduced methyltransferase activity, suggesting an important role of K656 in G-quadruplex RNA binding.

In short, we identified several key residues in human EZH2 that are important for RNA binding: F32, R34, D36 and K39 in the N-terminal helix, residues 489–494 (PRKKKR) in the CXC domain and

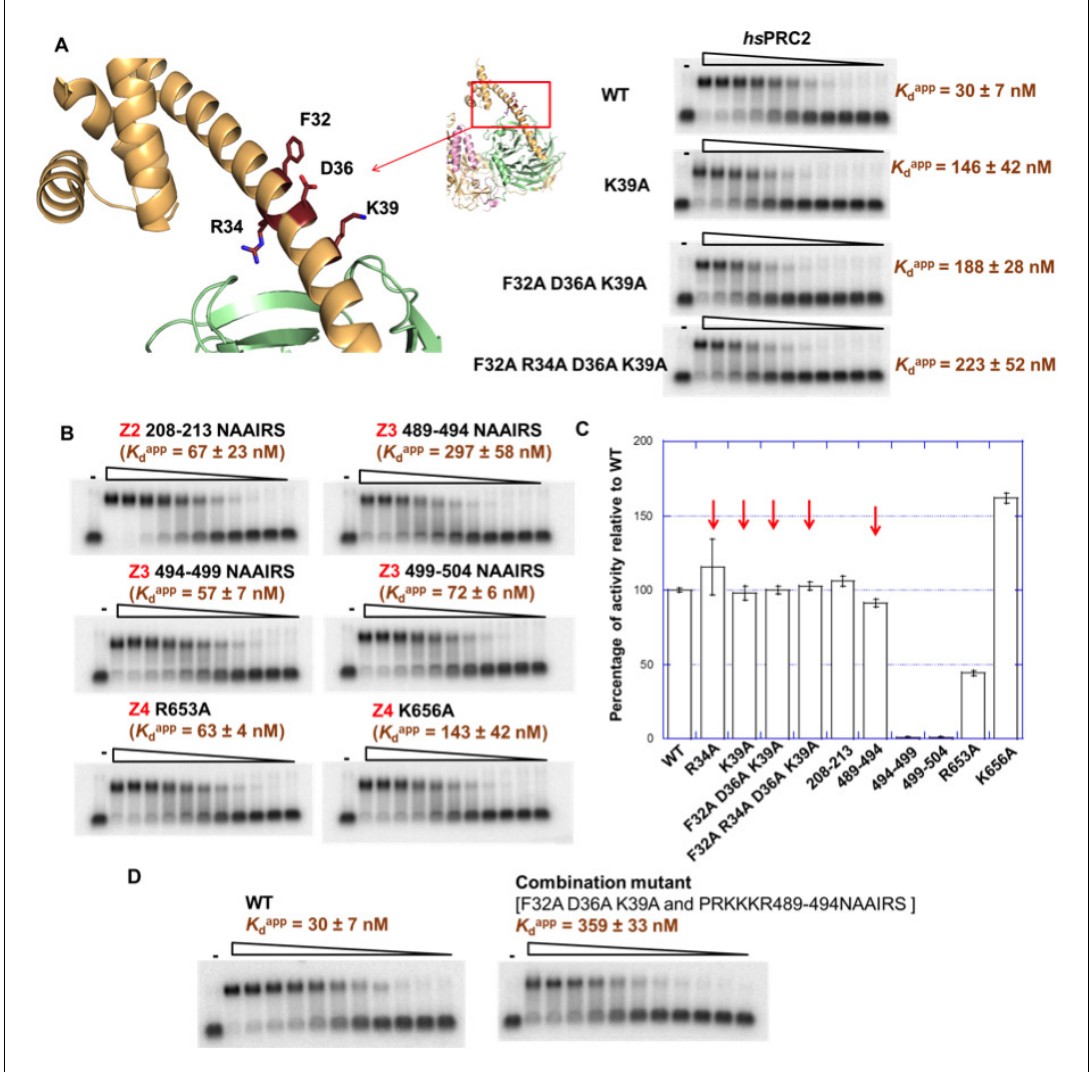

**Figure 6.** Mutagenesis results support RNA-binding roles of regions Z1, Z3 and Z4 in HDX data, and key residues are identified. (A) Identification of key residues in Z1. Positions of F32, R34, D36 and K39 in N-terminal helix of EZH2 are indicated in the *ac/hs*PRC2 crystal structure. K39A point mutant, triple mutant [F32A D36A K39A] and quadruple mutant [F32A R34A D36A K39A] were purified and tested for binding with (GGAA)$_{10}$ RNA, together with the wild type 3 m *hs*PRC2. All proteins were used at successive threefold dilutions starting at 2.5 μM concentration. (B) Mutagenesis in Z2, Z3 and Z4 leads to identification of important residues in Z3 and Z4. All proteins were used at successive threefold dilutions starting at 5 μM concentration, and $K_d^{app}$ values and errors are mean and standard derivation of three binding experiments performed at different days. (C) Histone methyltransferase activity assays for RNA-binding mutants, normalized to that of wild type and error calculated from three independent experiments. Red arrows indicate the mutants that disrupt RNA binding while maintaining normal catalytic activity. (D) Combining identified mutations into a single mutant [F32A D36A K39A and PRKKKR489-494NAAIRS] causes more severe binding defect for (GGAA)$_{10}$ RNA. Both proteins were used at successive threefold dilutions starting at 5 μM concentration, and $K_d^{app}$ values and errors are mean and standard derivation of three independent experiments.

DOI: https://doi.org/10.7554/eLife.31558.010

The following source data and figure supplements are available for figure 6:

**Source data 1.** Raw data in fRIP experiment.

DOI: https://doi.org/10.7554/eLife.31558.014

**Figure supplement 1.** A basic region (adjacent to and include Z3 in HDX-MS result) of EZH2 is important for RNA binding.

DOI: https://doi.org/10.7554/eLife.31558.011

**Figure supplement 2.** Mutations in the context of the 5 m *hs*PRC2 (holoenzyme) exhibit binding defect specifically for RNA relative to DNA.

DOI: https://doi.org/10.7554/eLife.31558.012

**Figure supplement 3.** Identified residues are important for (GGAA)$_{10}$ RNA interaction in vivo.

DOI: https://doi.org/10.7554/eLife.31558.013

K656 in the I-SET helix of EZH2. Importantly, H3 methylation activities were tested on all mutants and most had no change in catalytic activity (*Figure 6C*), except the K656A mutation resulted in an increase in activity (1.5 fold). Two important separation-of-function mutants were successfully made (the F32A, R34A, D36A and K39A quadruple mutant and the 489–494 NAAIRS substitution mutant). When we combined the key mutations, the RNA binding defect became more severe, with $K_d^{app}$ to $(GGAA)_{10}$ RNA increasing by around 12-fold compared to the wild-type 3 m *hs*PRC2 complex (*Figure 6D*). In addition, when we mutated these residues in the context of the holoenzyme - 5 m *hs*PRC2 (EZH2-SUZ12-EED-RBBP4-AEBP2), the mutant [F32A R34A D36A K39A and PRKKKR489-494NAAIRS] complex had no defects in assembly compared to the wild-type complex (*Figure 6—figure supplement 2A*), but exhibited a significant binding defect to $(GGAA)_{10}$ RNA in a fluorescent anisotropy binding assay (*Figure 6—figure supplement 2B*, left panel). Recently, we reported that *hs*PRC2 also binds to GC-rich dsDNA sequences, and RNA and DNA binding are mutually exclusive (*Wang et al., 2017a*). Importantly, the RNA-binding mutant still showed substantial binding to a GC-rich dsDNA (*Figure 6—figure supplement 2B*, right panel), providing a clear path for using it as a separation-of-function mutant for future in vivo experiments.

## Identified key residues of *hs*PRC2 are important for RNA binding in vivo

To verify the importance of the identified residues in vivo, a formaldehyde RNA immunoprecipitation (fRIP) experiment was performed in HEK293T cells using exogenously expressed FLAG-tagged *hs*EZH2 and a $(GGAA)_{10}$-MS2 RNA. The MS2 RNA stem loop is used to stabilize the $(GGAA)_{10}$ RNA, and the RNA expression has been verified previously using a Northern blot experiment (*Wang et al., 2017b*). Here, we co-expressed *hs*EZH2 wild type or mutants with the RNA, and then used fRIP to evaluate their interaction in cells. Three key RNA-binding defected mutants - [K39A], [F32A R34A D36A K39A], and [F32A R34A D36A K39A and PRKKKR489-494NAAIRS] - were expressed in HEK293T cells, and western blots analysis using a FLAG antibody indicated comparable expression levels of the wild type and mutant EZH2 proteins (*Figure 6—figure supplement 3A*). The K39A mutant and [F32A R34A D36A K39A] mutant exhibited moderate defects in interacting with the $(GGAA)_{10}$ RNA, while mutating all key residues identified in Z1 and adjacent to Z3 [F32A R34A D36A K39A and PRKKKR489-494NAAIRS] caused a significant reduction in RNA binding (0.27-fold relative to wild type EZH2, p=0.020, *Figure 6—figure supplement 3B*). These in vivo results support our in vitro finding of the importance of these residues for interacting with G-quadruplex RNA, and indicate that the RNA interaction is achieved by dispersed patches of amino acids in EZH2.

## Discussion

Numerous chromatin-binding proteins have been reported to interact with RNA in the last decade (*Hudson and Ortlund, 2014*; *G Hendrickson et al., 2016*), including histone modifiers, DNA methyl-transferases and transcription factors (reviewed in [*Long et al., 2017*]). However, many of these proteins do not contain an obvious RNA-binding motif such as an RRM (RNA recognition motif), and this has hindered critical tests of the function of RNA binding. In the few cases, when RNA-binding determinants were studied, RNA binding seemed to occur at multiple sites of the protein. For example, DNA methyltransferase 3 (DNMT3) has been shown to interact with an RNA at its catalytic domain in a structurally dependent way while two other RNAs can interact at an allosteric site outside the catalytic domain (*Holz-Schietinger and Reich, 2012*). Non-canonical RNA-binding sites might be widespread in such chromatin-binding proteins.

Here, we undertook a comprehensive study of the non-canonical RNA-binding regions of PRC2. First, we discovered that the RNA-binding specificity of PRC2 is conserved over vast evolutionary distance, from a fungus to human, which provides new evidence for its functional importance. We also identified several key RNA-binding residues of PRC2 from both of these species through a comprehensive analysis using mutagenesis and HDX-MS (*Figure 7*). The majority of these residues are located in EZH2, while *hs*EED also contains two regions - 290–308 (E1) and 338–350 (E2) - that are identified in HDX-MS and potentially important for interacting with RNA. The RNA-binding residues are dispersed across the protein surface, which may explain why identifying them by mutagenesis has been so challenging: if any single binding site is affected by mutation, the RNA can still bind to

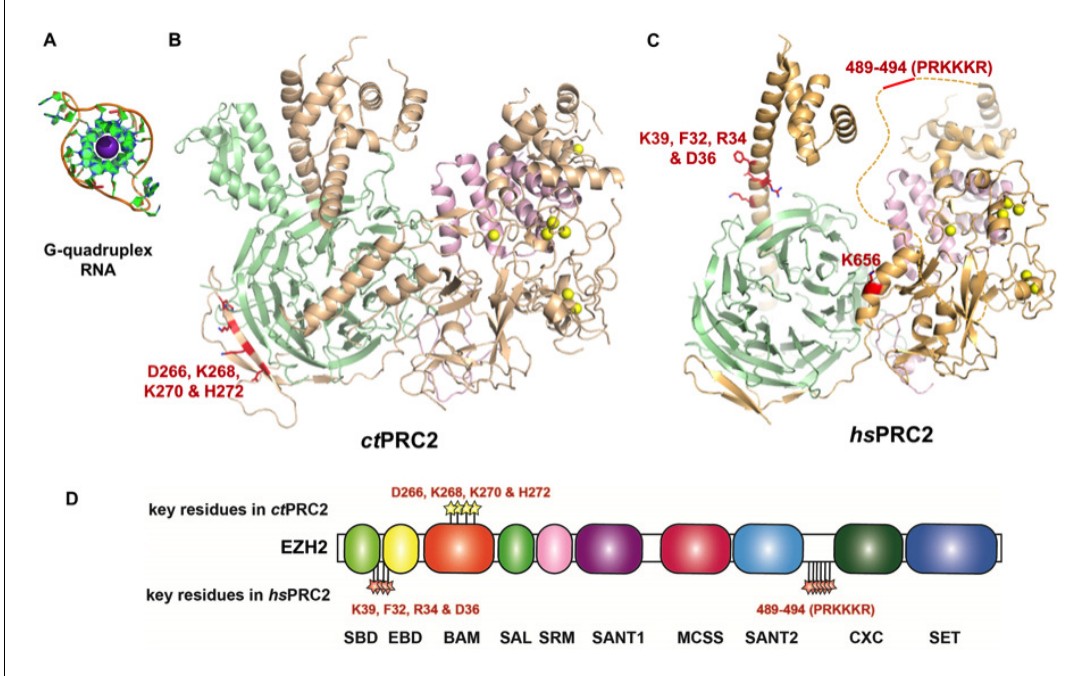

**Figure 7.** Summary of residues on EZH2 of *ct*PRC2 and *hs*PRC2 that are important for binding G-quadruplex RNA. (**A**) Crystal structure of a telomeric RNA quadruplex (PDB: 3IBK, sequence: UAGGGUUAGGGU) allows comparison of its size with that of PRC2 structures. Blue spheres in the center are potassium ions. (**B and C**) Identified key residues for RNA binding are shown in red. Subunits of PRC2 are colored differently. (**C**) Locations of the key residues for RNA binding are annotated along the polypeptide chain. Key residues in *ct*PRC2 are listed above the chain, while those in *hs*PRC2 are below the chain. Individual domains are color-coded as in *Figure 2A*. K656 mutation increases catalytic activity and is therefore omitted from this annotation map.

DOI: https://doi.org/10.7554/eLife.31558.015

another site(s) and the resultant change in $K_d^{app}$ is minimal. It will be interesting to see whether other epigenetic modifiers employ this non-canonical mode of RNA binding.

PRC2 from these two evolutionarily distant species (*C. thermophilum* and *H. sapiens*) share overall structural similarity and domain organization of the minimal catalytic unit (*Jiao and Liu, 2015*; *Brooun et al., 2016*; *Justin et al., 2016*), despite the less obvious sequence similarity outside the active site. To date, we have shown the specificity for G-rich RNA and G-quadruplex RNA for 3 m *ct*PRC2 (this study), 3 m *hs*PRC2 (this study) and 5 m *hs*PRC2 (*Wang et al., 2017b*). Given the limited sequence homology between *ct*PRC2 and *hs*PRC2, it is striking that the RNA-binding affinity and specificity is evolutionarily conserved over a vast evolutionary distance, inferring that RNA binding of PRC2 could have critical functions in eukaryotes. The best-established function of PRC2-RNA interaction at this point is inhibition of PRC2's histone methyltransferase activity (*Cifuentes-Rojas et al., 2014*; *Kaneko et al., 2014*; *Beltran et al., 2016*; *Wang et al., 2017a*).

This study also reveals that the 3 m PRC2 complex contains elements sufficient for the RNA-binding specificity. Surprisingly, almost none of the identified key RNA-binding residues are conserved between *ct*PRC2 and *hs*PRC2, and the domains important for RNA interaction are also different. It is possible that *ct*PRC2 and *hs*PRC2 found different paths to achieve the specificity for G-quadruplex RNA, an example of convergent evolution. It is interesting that *ct*EED contains an extra domain that is adjacent to the N-terminal helix of EZH2 (*Figure 7*), and this extra domain might prevent RNA interaction with the N-terminal helix (Z1 in *hs*EZH2) due to steric hindrance. Thus *ct*PRC2 could not use the N-terminal helix for RNA interaction as *hs*PRC2 does, and instead needs an alternative site to achieve the RNA-binding specificity.

Notably, missense mutations of some of the *hs*EZH2 residues critical for RNA binding, including R34, K39 and R494, have been identified in cancer patients (R34L, K39E and R494S mutations, https://portal.gdc.cancer.gov). However, it remains unclear what physiological consequences these mutations have in cancer, because each of these heterogeneous mutations was identified in one out

of hundreds of patients investigated and in different type of cancers (R34L and K39E were both identified in lung adenocarcinoma patients). Given that RNA binding inhibits PRC2's ability to act as a histone methyltransferase, these mutant PRC2 proteins might be expected to have increased activity in vivo, a possibility that remains to be investigated. Thus, in addition to EZH2 overexpression, mutation of RNA-binding residues may also drive PRC2 gain-of-function in cancer.

Previously, residues 342–368 (RBD1) in mouse EZH2 were reported to be important for RNA interaction (*Kaneko et al., 2010*), and this sequence is conserved in *hs*EZH2. Our study was unable to support this finding using three independent experiments: [1] deletion of residues 342–368 in a purified mouse 4 m PRC2 (EZH2-EED-SUZ12-RBBP4) results in no change in binding to a RepA noncoding RNA (*Figure 4—figure supplement 1*); [2] deletion of residues 342–368 in a 3 m *hs*PRC2 complex (EZH2-EED-VEFS(SUZ12)) doesn't affect binding to a $(GGAA)_{10}$ G-quadruplex RNA (*Figure 4B*, middle left); [3] deletion of the unstructured region in *ac*EZH2, corresponding to region 342–368 of mEZH2 in the context of the hybrid *ac/hs*PRC2, does not affect $(GGAA)_{10}$ RNA binding or the crosslinking to a 24 nt G-quadruplex RNA. Although it is possible this region is important for interaction with some other RNA that was not tested in this study, this may be unlikely given that different RNAs compete for binding and show similar RNA-protein crosslinking patterns (*Wang et al., 2017b*). It also seems possible that this portion of EZH2 might contribute to RNA binding, perhaps indirectly, in some form of PRC2 not studied here. Our study also did not test the phosphorylation of T345, which could be important for noncoding RNA interaction as indicated in the previous study (*Kaneko et al., 2010*).

HDX-MS is used in this study as an important and powerful discovery tool to study protein-RNA interactions, and it also provides additional information regarding changes in flexibility that can be sometimes caused by allosteric change rather than direct interaction. Other groups have previously used this approach to investigate protein-RNA interactions (*Lísal et al., 2005*; *Zheng et al., 2015*), as it provides extended sequence coverage and resolution compared to other residue-specific methods including *N*-hydroxy-succinimide (NHS)-biotin labeling of surface lysine residues (*Kvaratskhelia and Grice, 2008*).

The results in this study suggest a potential essential role of PRC2-RNA interaction through evolution, and identification of the key residues opens the door to uncover the importance of PRC2-RNA interaction in vivo. The critical RNA-binding regions identified in this study when mutated retain substantial histone methyltransferase activity and DNA-binding affinity, and therefore provide a clear path for making separation-of-function mutants to study biological functions of PRC2-RNA interaction in vitro and in vivo. It will be valuable to determine the biological consequences of introducing an RNA-binding defective mutant of PRC2 by investigating changes in global RNA binding pattern, PRC2 chromatin occupancy and genome-wide gene expression, and the findings in this study open the door to study physiological functions of PRC2-RNA interaction including how lncRNAs recruit or evict PRC2 and regulate its activities. Recently, G-quadruplex-forming RNA sequences have been proposed to be largely unfolded in vivo (*Guo and Bartel, 2016*), and it would be interesting to study whether PRC2 can bind and stabilize transiently folded quadruplexes. In addition, the molecular details of how key PRC2 residues cooperate and contribute to RNA interaction is still an open question, and structural analysis including NMR, crystallography and electron microscopy could be used to understand the details of the interaction, which could serve as a guide to study protein-RNA interaction of other chromatin-binding proteins.

## Materials and methods

### Expression and purification of *ct*PRC2 complexes

Wild-type 3 m *ct*PRC2 was purified as previously described (*Jiao and Liu, 2015*). Two polypeptides – an EZH2-VEFS peptide with an N-terminal 2XProtein A tag and a 6XHis tag and an EED peptide with an N-terminal twin-strep tag (IBA Lifesciences) – were expressed in *S. cerevisiae* strain CB010, and intact *ct*PRC2 was purified through tandem purification using IgG resin followed by Strep-Tactin resin. Regulatory moiety (N-terminal His tag on EZH2), minimal RNA-binding complex (including trimming mutants, N-terminal His tag on EZH2), EED (N-terminal His tag) and CXC-SET (N-terminal His tag) were expressed in *Escherichia coli* strain Rosetta 2 (Novagen). Cell pellets were resuspended in lysis buffer containing 100 mM Tris-HCl (pH 8.5), 300 mM NaCl, 1 mM dithiothreitol (DTT), 10%

glycerol, 0.5 mM phenylmethylsulfonyl fluoride, and protease inhibitors, and lysed using sonication. Lysate was then cleared using centrifugation and incubated with Ni-NTA Agarose resin (Qiagen) for 2 hr. Resin was washed stepwise by 50-fold resin volume of buffer A [50 mM Tris-HCl (pH 8.0), 500 mM NaCl, 1 mM DTT, 10% glycerol, and 0.1% NP40], 50-fold resin volume of buffer B [50 mM Tris-HCl (pH 8.0), 1 M NaCl, 1 mM DTT, and 10% glycerol), 20-fold resin volume of buffer C [50 mM Tris-HCl (pH 8.0), 100 mM NaCl, 1 mM DTT, and 10% glycerol], 20-fold resin volume of buffer C containing 10 mM imidazole, and finally eluted with buffer C containing 250 mM imidazole. Elutions were concentrated and loaded to Superdex 75 size-exclusion chromatography column with running buffer containing 20 mM Tris-HCl (pH 8.0), 100 mM NaCl, and 2 mM DTT. Peak fractions were pooled, analyzed with denaturing gel and concentrated for downstream analysis. For point mutations in the context of the intact 3 m $ct$PRC2, mutants and wild type proteins were expressed in $S. cerevisiae$ strain CB010 and purified using Ni-NTA agarose resin as described above followed by resolving using Superdex 200 size-exclusion chromatography column.

SANT2L-VEFS complex was expressed by co-transforming pSumo-$ct$SANT2L(EZH2, residues 490–640, His$_6$-Sumo tag) and pGex4T1-$ct$VEFS(SUZ12, residues 572–691, GST-thrombin tag) plasmids in Rosetta 2 (DE3) competent cells, and VEFS protein alone was expressed by only transforming the pGex4T1-$ct$VEFS plasmid. When $OD_{600}$ reached 0.6, protein expression was induced by 0.5 mM IPTG at 20°C for 20 hr. Cells were lysed by sonication in lysis buffer containing 50 mM Tris-HCl, pH 8.0, 150 mM NaCl, 10% glycerol, 2.5 mM DTT, and 1 mM PMSF. Cell lysate was clarified by centrifugation and the supernatant was mixed with Glutathione Agarose (Pierce) for batch binding at 4°C for 2 hr. The glutathione agarose was washed by wash buffer 1 containing 50 mM Tris-HCl, pH 8.0, 150 mM NaCl, 10% glycerol, 2.5 mM DTT, 0.1% NP-40, and then by wash buffer 2 containing 50 mM Tris-HCl, pH 8.0, 150 mM NaCl, 10% Glycerol, 2.5 mM DTT. Sumo protease was added to cleave the His$_6$-Sumo-$ct$SANT2L (EZH2) fusion protein overnight, after which the glutathione agarose was washed by wash buffer 2 (this step was omitted when purifying VEFS alone). Thrombin was then added to cleave the GST-$ct$VEFS (SUZ12) protein overnight, and the binary complex was eluted by elution buffer containing 50 mM Tris-HCl, pH 8.0, 150 mM NaCl, 2.5 mM DTT. The SANT2L-VEFS complex or VEFS alone was further purified on a Superdex 75 gel filtration column in gel filtration buffer containing 20 mM Tris-HCl, pH 8.0, 100 mM NaCl and 2 mM DTT.

## Electrophoretic mobility shift assay

Binding assays were conducted similarly as previously described (*Wang et al., 2017b*). All 40-mer RNAs were synthesized by Dharmacon and then end-labeled with gamma-$^{32}$P-ATP. RNA was diluted in TE pH 7.5, heated for 10 min at 95°C, and snap-cooled on ice for 3–5 min. RNA was then allowed to fold for 30 min at 37°C in binding buffer (50 mM Tris-HCl pH 7.5 at 25°C, 100 mM KCl, 2.5 mM $MgCl_2$, 0.1 mM $ZnCl_2$, 2 mM 2-mercaptoethanol, 0.1 mg/ml bovine serum albumin, 0.1 mg/ml fragmented yeast tRNA (Sigma cat # R5636), 5% v/v glycerol). Protein samples were diluted with binding buffer, and then mixed with refolded radiolabeled RNA (1000 cpm/lane, 1:1 ratio) at 30°C for 30 min. Sample was loaded to non-denaturing 0.7% agarose gel (SeaKem GTG Agarose, Fisher Scientific cat # BMA 50070) buffered with 1XTBE at 4°C and resolved at 66 V for 90 min in a 4°C cold room. Gels were vacuum dried for 60 min at 80°C on a Hybond N + membrane (Amersham, Fisher Scientific 45-000-927) and a sheet of Whatman 3 mm chromatography paper. Dried gels were exposed to phosphorimaging plates and signal acquisition was performed using a Typhoon Trio phosphorimager (GE Healthcare). For PRC2-RNA binding in LiCl buffer, 100 mM KCl was substituted with 100 mM LiCl in binding buffer.

## Expression and purification of *hs*PRC2 complexes

Two expression systems were used for purification of wild type and mutant $hs$PRC2.

For *S. cerevisiae* expression system (proteins in *Figure 4B*), a DNA fragment corresponding to the sequence 2 × ProA SSGENLYFQSNHHHHHHA, was appended N-terminal to the DNA encoding EZH2 (residues 1 to 746)-LVPRGS-VEFS (SUZ12, residues 543 to 695), and was sub-cloned into the p416GAL1 vector. The EZH2-VEFS(SUZ12) construct was co-transformed with full length EED (residues 1–441) in a modified p416GAL1 vector (LEU marker) into *S. cerevisiae* CB010 strain (*MATa pep4::HIS3 prb1::LEU2 prc::HISG can1 ade2 trp1ura3 his3 leu2-3,112*) and transformants were selected on synthetic complete media lacking uracil and leucine. All the constructs harboring

mutations were generated with the site-directed mutagenesis strategy. Cell starters were grown in synthetic complete drop-out medium without uracil and leucine (Sc-Ura-Leu) supplemented with 2% raffinose at 30°C for 24 hr. Protein expression was induced by inoculating the cell starters into the Sc-Ura-Leu medium supplemented with 2% galactose at 30°C for an additional 24 hr. Protein purification for human PRC2 followed a similar procedure as detailed previously (*Jiao and Liu, 2015*). Briefly, protein was purified by IgG sepharose followed by overnight cleavage with TEV protease and a final gel filtration step on Superdex 200 in 20 mM Tris-HCl, pH 8.0, 100 mM NaCl, and 2.5 mM DTT.

For insect cell expression system, expression of the 3 m *hs*PRC2 wild type complex and its mutants was conducted as previous described using multi-ORF pFastBac Dual vector allowing expression of full length human EZH2, EED (81-441) and SUZ12-VEFS (545-726; S583D) (*Brooun et al., 2016*). Mutants of EZH2 subunit of *hs*PRC2 were generated at GeneScript using proprietary site-specific mutagenesis protocols. Viruses were generated using standard Bac-to-Bac viral generation protocols (Life Technologies) and amplified to high-titer passage two (P2) stocks. Protein overexpression was conducted in exponentially growing Sf9 or Sf21 insect cells (depending on the optimum cell line for specific constructs) infected at $2 \times 10^6$ with P2 viral stock at MOI = 1. All wild-type and mutant *hs*PRC2 were purified identically. They were purified from cell lysate using Flag-affinity chromatography. Cells were lyzed in 50 mM Tris 8.0, 200 mM NaCl, 5% glycerol, 0.25 mM TCEP supplemented with EDTA-free protease inhibitor cocktail (Roche). 1.5 ml of lysis buffer was added per 1 g of frozen biomass. The clarified lysate was obtained by centrifugation of cell lysate at 10,000 g for 1 hr at 4°C. 5 ml of Anti-FLAG M2 Agarose (Sigma) was added per 5 L of biomass and incubated for 3 hr at 4°C (batch binding). Flag resin bound to PRC2 was washed with 20 column volumes (CV) of 50 mM Tris pH 8.0, 200 mM NaCl, 5% glycerol, 0.25 mM TCEP followed by elution of PRC2 using 3 CV of 50 mM Tris 8.0, 200 mM NaCl, 5% glycerol, 0.25 mM TCEP supplemented with 200 g ml$^{-1}$ of FLAG Peptide (DYKDDDDK). As expected, single step FLAG purification allowed stoichiometric capture of all three desired subunits of PRC2. PRC2 was further purified using S200 26/600 column (GE Healthcare) pre-equilibrated with 2 CV of 25 mM Tris pH 8.0, 200 mM NaCl, 5% glycerol, 0.5 mM TCEP buffer. Peak fractions containing *hs*PRC2 monomer were concentrated to 2–20 mg ml$^{-1}$, flash frozen in small aliquots using liquid nitrogen and stored at −80°C.

4 m and 5 m *hs*PRC2 complexes were expressed and purified as previously described (*Davidovich et al., 2013*).

## Hydrogen deuterium exchange mass spectrometry (HDX-MS)

Insect cell-expressed 3 m *hs*PRC2 samples (5 μM) were prepared with and without the addition of RNA (40 μM) in stock buffer (50 mM Tris, pH 7.5, 100 mM KCl, 1 mM TCEP, 5% glycerol) for differential HDX-MS analysis. Deuterium exchange was conducted on a temperature-controlled HDX2 autosampler (Leap Technologies). Aliquots of sample (4 μL) were mixed (1:4) with deuterium exchange buffer (D$_2$O 25 mM Tris, 150 mM NaCl) at 4°C. Exchange was conducted across five time points (10 s, 60 s, 360 s, 3600 s, 28,800 s) and run in replicates (triplicate/quadruplicate). After a defined exchange time, the sample was transferred by a chilled syringe, and quenched/denatured at 1°C with addition of 20 μL of chilled quench buffer (3.2 M guanidine hydrochloride, 0.8% formic acid). Quenched samples were injected into the chiller box, which housed the sample loop, protease column and trap/analytical columns at 3°C. Blank injections were inserted between every sample to minimize potential carryover.

A three pump scheme was employed for peptide detection using Vanquish UPLC pumps (Dionex). A loading pump running 0.1% formic acid at 200 μl/min carried sample sequentially across the pepsin/protease XIII immobilized column (NovaBio Assay) and 2.1 × 5 mm CSH C18 trap column (Waters) for 2 min. Peptides were desalted for an additional 30 s, while the protease column was back-flushed with 0.05% TFA. Peptides were subsequently eluted/separated across a Kinetex C8 2.1 × 50 mm analytical column via a 5.7-min gradient of mobile phase B (8%–37% ACN, 0.1% formic) employing a binary pump gradient with mobile phase A (0.1% formic acid) and mobile phase B (acetonitrile 0.1% formic acid) at 150 μL/min.

Data acquisition and processing involved multiple software platforms. Mass spectra were acquired on a Thermo Fusion-Lumos mass spectrometer running XCalibur 2.1 across a scan range of 375–1300 m/z. Orbitrap resolution was set at 60,000 for charge state selection of +2 to + 5 peptides. Peptide fragmentation for peptide identification employed both HCD and EThcd tandem MS2

acquisition. Electrospray source settings were set at high-gas flow (27 sheath, nine aux) and temperature (150℃) to handle the 150 µl/min LC flow rate, while minimizing potential deuterium back-exchange. Peptide pools were generated in Thermo Proteome Discoverer 2.1 with a directed search against the protein sequence, using Sequest HT search and a fixed value PSM validator (0.05 Delta Cn). Tolerances were set at five ppm mass accuracy with 0.6 Da fragment. Deuterium exchange was determined with HDExaminer software. For peptide deuterium uptake, the first two residues and prolines were excluded from calculations.

## Histone methylation activity assays

For *ct*PRC2, 600 nM of purified wild type or mutant protein was incubated with 12 µM $^{14}$C-labelled SAM (PerkinElmer) and 2.4 µM histone H3.1 proteins (NEB, M2503S) for 30 min in a 1X reaction buffer containing 50 mM Tris-HCl, pH 7.5, 100 mM KCl, 5 mM MgCl$_2$, 0.5 mM ZnCl$_2$, 0.1 mM CaCl$_2$, 2 mM 2-mercaptoethanol. Reactions were then loaded on a Nupage 4–12% Bis-Tris gel (Life Technologies). Gel electrophoresis was carried out for 60 min at 150 Volts, and the gel was vacuum dried at 80℃ for 30 min on Whatman 3 mm chromatography paper. Dried gels were exposed to phosphorimaging plates, and signal acquisition was performed with a Typhoon Trio phosphorimager (GE Healthcare).

For insect cell-expressed *hs*PRC2 and mutants, methyltransferase activity was measured using a radioactive filter-binding assay as previously described (**Brooun et al., 2016**). For *S. cerevisiae*-expressed *hs*PRC2 wild type and R34A mutant, 500 ng of proteins were mixed with 5 µM of histone H3K27me0 peptide (residues 21–44, AnaSpec catalogue number AS-64641) and 4 µL of $^3$H SAM (1 mCi stock, specific activity of 78–85 Ci/mmol; Perkin Elmer) in a final volume of 80 µL (final concentration of about 0.5 µM $^3$H SAM per reaction), and incubated at 30℃ in a buffer containing 25 mM Tris-HCl, pH 8.0, 10 mM NaCl, 1 mM EDTA, 2.5 mM MgCl$_2$, and 5 mM DTT. After 30 min, 10 µL of the reaction mixture was transferred to a tube to stop the reaction with 2 mM of unlabeled SAM, spotted onto phosphocellulose filters (Reaction Biology Corporation), and allowed to dry. Filters were washed five times with 50 mL of 50 mM Na$_2$CO$_3$/NaHCO$_3$, pH 9.0 and once with 30 mL of 100% acetone and dried. Filters were immersed in 4 mL of scintillation fluid and counted with a scintillation counter set to output readings as disintegrations per minute (DPM). All experiments were performed in triplicate, and activities of mutants were normalized to that of the wild-type protein.

## UV crosslinking of RNA-PRC2 complexes

4-thio-U incorporated RNA (sequence: UUUGGGU(4-thio-U)UGGGUUGGGUUGGGUU) was synthesized by Dharmacon and then end- labeled with gamma-$^{32}$P-ATP. Trace amounts of hot RNA were refolded (similar as above in EMSA assays, except BSA and tRNA were omitted from the buffer) and then incubated with 200 nM *ac/hs*PRC2 protein for 30 min at 30℃. Each sample was moved to siliconized glass coverslips on ice, placed in a Stratalinker with 365 nm bulbs, and crosslinked for 30 min. SDS loading dye was added and samples were then resolved on a Nupage 4–12% Bis-Tris gel (Life Technologies) for 60 min at 180 V. Gels were vacuum dried at 80℃ for 30 min on Whatman 3 mm chromatography paper. Dried gels were exposed to phosphorimaging plates and signal acquisition was performed using a Typhoon Trio phosphorimager (GE Healthcare).

## Fluorescence anisotropy binding assays

Fluorophore-attached nucleic acids were commercially synthesized with a 3'((GGAA)$_{10}$ RNA) or 5'(dsDNA) Alexa-488 fluorescent dye. (GGAA)$_{10}$ RNA was refolded and bound to various concentrations of PRC2 complexes the same way as described in the EMSA binding assay, except yeast tRNA was omitted in the binding buffer. (GC)$_{30}$ dsDNA was incubated with PRC2 complexes in a low-salt binding buffer (10-fold less KCl and MgCl$_2$) to facilitate binding (because DNA binding is much weaker compared to RNA binding). After 30 min of incubation, fluorescent polarizations were measured in a 384-well plate using a Synergy two multi-mode plate reader (BioTek).

## Formaldehyde RNA immunoprecipitation (fRIP)

HEK293T cells grown in DMEM medium were transfected with three plasmids pCDNA3.1-3XFLAG-Halo-EZH2 (WT or mutants), P_MU6-(GGAA)10-MS2 and pMS2-eGFP (Addgene:#27121) using Lipofectamine 2000 reagent (Life Technologies). At 48 hr post transfection, media were aspirated and

cells were fixed and crosslinked by adding 1XPBS containing 0.1% formaldehyde. After 10 min of incubation at room temperature, crosslinking was quenched by addition of 125 mM glycine. Cells were harvested by scrapping using cell lifters (Corning), washed once in cold 1XPBS, and stored at −80°C. Cell lysis and immunoprecipitation protocols were adapted from (*G Hendrickson et al., 2016*). Cells harvested from a 10-cm dish were re-suspended in 0.5 mL of RIPA lysis buffer (50 mM Tris, pH 8.0, 150 mM KCl, 0.1% SDS, 1% Triton-X100, 5 mM EDTA, 0.5% sodium deoxycholate, 0.5 mM DTT, 100 U/mL RNaseOUT (Life Technologies) and protease inhibitor cocktail (ThermoFisher-Pierce 88266)) and incubated on ice for 10 min. Cells were then sonicated for 15 min with a Bioruptor UCD-200 (Diagenode) with 30 s pulses at maximum power, and lysate was cleared by centrifuging at 16,300 g for 10 min. Supernatant was collected and diluted by adding equal volume (500 μL) of fRIP binding/wash buffer (25 mM Tris, pH7.5, 150 mM KCl, 5 mM EDTA, 0.5% NP-40 substitute (Sigma 74385), 0.5 mM DTT, 100 U/mL RNaseOUT and protease inhibitor cocktail). 25 μL of this diluted lysate was saved as the input sample, and 75 μL was used for immunoprecipitation by mixing with 15 μL packed anti-FLAG M2 affinity beads (Sigma A2220, pre-equilibrated in fRIP binding/wash buffer) in 125 μL fRIP binding/wash buffer (200 μL final volume). The mixture was rotated at 4°C for 2 hr and then washed four times with the fRIP binding/wash buffer (centrifuging at 100 g for 5 min for each wash). After the last wash, the beads were re-suspended in 67 μL ddH$_2$O and 33 μL 3X reverse-crosslinking buffer (3XPBS, 6% N-lauroyl sarcosine, 30 mM EDTA, 15 mM DTT, 3 mg/mL Proteinase K, 300 U/mL RNaseOUT), and the 25 μL input sample was mixed with 33 μL 3X reverse-crosslinking buffer and 42 μL ddH$_2$O. Protein degradation and reverse-crosslinking were performed for 1 hr at 42°C and 1 hr at 55°C. 1 mL Trizol (Life Technologies, Carlsbad, CA) was then added and RNA extraction was performed following manufacturer's instructions. Extracted RNA was treated with RQ1 RNase-free DNase (Promega M6101) for 1 hr at 37°C, and then extracted using phenol: chloroform: isoamyl alcohol followed by ethanol precipitation. All precipitation steps were facilitated by addition of 2 μg glycogen per sample. RNA pellet was re-suspended in 20 μL ddH$_2$O, and 2 μL was used for making cDNA using SuperScript III (Life Technologies), 1 μM random hexamer and 500 μM dNTPs. qPCR was performed using SYBR Select Master Mix (ThermoFisher) and primers to the MS2 region as previous described (*Wang et al., 2017b*). $\text{Fraction of pulldown} = \frac{2^{\text{Ct(input)}-\text{Ct(IP)}}}{3}$. Results from three biological replicates were averaged and errors are the standard deviations.

Western blot experiments were used to compare expression levels of wild type and mutant *hs*EZH2. Transfections of the three plasmids were performed similar as described above, except at a smaller scale in 6-well plates. 24 hr after transfection, media were aspirated and cells were harvested by adding 200 μL 1X NuPAGE LDS Sample Buffer (Invitrogen NP0007) and 2 μL Benzonase Nuclease (Sigma E1014) per well. Lysate was incubated at 37°C for 30 min to completely digest nucleic acids. Samples (2.5 μL and 0.625 μL) of lysate were resolved in a NuPAGE 4–12% bis-tris protein gel (ThermoFisher) and then transferred to a Hybond ECL membrane (GE# RPN78D) in 1X transfer buffer (25 mM tris base, 192 mM glycine, 0.1% SDS and 20% methanol) at 0.5 Amps constant for 1 hr. The membrane was blocked with 10 mL StartingBlock T20 (PBS) Blocking buffer (Thermo # 37539) at room temperature for 30 min, incubated with 2 μL anti-FLAG M2-Peroxidase (HRP) antibody (Sigma A8592) in 10 mL blocking buffer for 1 hr, washed 3 times with 10 mL 1XPBS with 0.05% tween20 and once with 10 mL 1XPBS, and developed using SuperSignal West Pico Chemiluminescent Substrate kit (Thermo Scientific 34080).

## Acknowledgements

We thank members of the Cech lab, Oncology Structural Biology and Protein Science group, and the Liu lab for useful conversations. We also thank Daniel T Youmans for technical advice and reagents for the RNA immunoprecipitation experiments. TRC is an investigator of the HHMI, which supported this research. This research was supported by Welch Foundation research grant I-1790, CPRIT research grant R1119, Rita Allen Foundation research grant, UT Southwestern Medical Center Endowed Scholar fund, and NIH grants GM114576 and GM121662 to XL XL is a W W Caruth, Jr. Scholar in Biomedical Research. This research also received support from the Cecil H and Ida Green Center Training Program in Reproductive Biology Sciences Research.

# Additional information

## Competing interests

Ben Bolanos, Wei Liu, Karen A Maegley, Ketan S Gajiwala, Alexei Brooun: The research was funded by Pfizer, Inc., where this author was employed at the time the study was conducted. The author declares no other competing financial interests. Thomas R Cech: T.R.C. is on the board of directors of Merck. The other authors declare that no competing interests exist.

## Funding

| Funder | Grant reference number | Author |
|---|---|---|
| Pfizer | | Alexei Brooun |
| Howard Hughes Medical Institute | | Thomas R Cech |
| Welch Foundation | research grant I-1790 | Xin Liu |
| Cancer Prevention and Research Institute of Texas | research grant R1119 | Xin Liu |
| Rita Allen Foundation | | Xin Liu |
| National Institute of General Medical Sciences | GM114576 | Xin Liu |
| National Institute of General Medical Sciences | GM121662 | Xin Liu |
| University of Texas Southwestern Medical Center | Endowed Scholar Fund | Xin Liu |
| The Cecil H. and Ida M. Green Center | | Xin Liu |

The funders had no role in study design, data collection and interpretation, or the decision to submit the work for publication.

## Author contributions

Yicheng Long, Conceptualization, Data curation, Formal analysis, Validation, Investigation, Visualization, Methodology, Writing—original draft, Writing—review and editing; Ben Bolanos, Conceptualization, Data curation, Formal analysis, Investigation, Visualization, Methodology, Writing—original draft, Writing—review and editing; Lihu Gong, Formal analysis, Investigation, Writing—original draft; Wei Liu, Karen A Maegley, Data curation, Formal analysis, Investigation, Methodology; Karen J Goodrich, Data curation, Formal analysis, Validation, Investigation; Xin Yang, Siming Chen, Investigation; Anne R Gooding, Data curation, Formal analysis, Validation, Investigation, Visualization, Methodology; Ketan S Gajiwala, Conceptualization, Visualization, Writing—review and editing; Alexei Brooun, Conceptualization, Data curation, Supervision, Methodology, Writing—original draft, Writing—review and editing; Thomas R Cech, Conceptualization, Supervision, Funding acquisition, Methodology, Writing—review and editing; Xin Liu, Conceptualization, Formal analysis, Supervision, Funding acquisition, Methodology, Writing—review and editing

## Author ORCIDs

Yicheng Long http://orcid.org/0000-0003-3578-5197
Thomas R Cech https://orcid.org/0000-0001-7338-3389
Xin Liu https://orcid.org/0000-0002-5646-423X

## Decision letter and Author response

Decision letter https://doi.org/10.7554/eLife.31558.020
Author response https://doi.org/10.7554/eLife.31558.021

## Additional files

### Supplementary files

• Supplementary file 1. List of all proteins used in this study
DOI: https://doi.org/10.7554/eLife.31558.016

• Supplementary file 2. Mutations in *ct*PRC2 that had <3 fold effect on affinity for (G4A4)4 RNA
DOI: https://doi.org/10.7554/eLife.31558.017

• Transparent reporting form
DOI: https://doi.org/10.7554/eLife.31558.018

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
