## [Decision Letter]

Thank you for submitting your article "Conserved RNA Binding Specificity of Polycomb Repressive Complex 2 Is Achieved by Dispersed Amino Acid Patches in EZH2" for consideration by *eLife*. Your article has been reviewed by two peer reviewers, and the evaluation has been overseen by a Reviewing Editor and John Kuriyan as the Senior Editor. The reviewers have opted to remain anonymous.

All of the reviewers and Editors are positive about the study and find it to be a major contribution to the polycomb field. Therefore, we are ready to move forward with the acceptance of the manuscript for publication. Prior to formal acceptance, we wanted to give you the opportunity to textually address the minor comments/suggestions made by expert reviewers. We have included the reviewers comments at the bottom of this letter for you and look forward to receiving the revised version of your study.

*Reviewer #1:*

PRC2 binds RNA molecules promiscuously, and the authors of this manuscript have recently reported that this occurs with a preference for G-rich sequences and G-quadruplex structures (Wang et al., 2017).

Here, they show that RNA binding preference is the same for evolutionarily distant ctPRC2 as for human PRC2, and they characterize the binding interface on PRC2 for both species. Experiments and interpretations of data are thorough and of high quality, and identification of binding interfaces are convincing.

The physiological relevance of promiscuous RNA binding by PRC2 is not established in the literature. The authors have made great efforts to identify separation-of-function mutants that can aid in addressing the physiological function of RNA binding, but they have not applied their mutants towards such investigations. The authors report a mutant with reduced RNA binding (to approximately 25%), and it is not clear, if this reduction is sufficient to facilitate investigation into physiological relevance.

In summary, the technical quality of this study is very high and the authors address a very interesting question. Unfortunately, the authors have not used their RNA-binding PRC2 mutants to address the physiological role of RNA-binding for PRC2 function. Thus, it appears that the authors have stopped half-way, and publication in *eLife* cannot be recommended.

*Reviewer #2:*

Long et al. investigated the molecular basis of RNA-binding by PRC2. Using the previously crystallized conserved catalytic core comprising EED, EZH2 and a short fragment of SUZ12 from both Chaetomium and human PRC2, they perform rigorous biochemical binding assays combined with HDX-MS mapping studies to identify residues that are important for RNA interaction. Binding studies with PRC2 versions containing mutations in these residues confirm their importance for RNA interaction. Surprisingly, the location of Chaetomium and human PRC2 amino acid residues required for RNA binding (i.e. G-quadruplex RNA) are not conserved, and they are also not located on the same surfaces in the two complexes.

This is a very nice study of excellent technical quality. The paper definitely merits to be published in *eLife*. There is a vast amount of literature accompanied by considerable hype about RNA-binding by PRC2. Yet, as the authors state, to date it still is completely unclear whether RNA binding by PRC2 is of any physiological significance at all. The regions identified in this study will allow performing the relevant mutational analyses (i.e. EZH2 knock-in mutations in mice) to clarify this issue. Such experiments are beyond the scope of this study but will hopefully be done in the near future.

I only have a few minor comments of editorial nature that the authors should address.

1) Figure 3. This is radioactive signal and so it represents H3 molecules that were mono-, di- or tri-methylated in the reaction. The band should therefore be labelled as H3K27me and not as H3K27me3. Alternatively, the authors should provide WB data with an H3K27me3-specific antibody.

2) Figure 7 is not a very strong dataset. The entire figure could go into the supplement.

3) Figure 8 is a nice summary. So it would be helpful to clearly indicate in the figure that panel B is C.t. PRC2 and C is H.s. PRC2. It may also be helpful to include a C.t and H.s. EZH2 domain architecture diagram (i.e. like in Figure 2), indicating the location of the relevant residues along the polypeptide chain.

4) Discussion, first paragraph: I don't think that the term "RNA binding domain" is helpful in this case. The term "domain" should be reserved for a portion of a protein with a defined fold. Here, the authors show that the interaction of PRC2 with RNA occurs via positively charged amino acid residues that are dispersed over a large protein surface and are not even in a conserved location on the complex when comparing C.t. and H.s. PRC2.

5) The authors state that "missense mutations of some the H.s. EZH2 residues critical for RNA binding have been identified in cancer patients." The https://portal.gdc.cancer.gov website lists about 150 different amino acid changes that have been identified in EZH2 (which is a 746 amino acid protein) and almost all of them – including the residues mentioned in the text – were identified a single time and in completely different tumours. So it is not clear how meaningful these specific particular mutations are. Moreover, I assume that these are all heterozygous mutations in tumours where the homologous chromosome carries the EZH2 wild-type allele. Please clarify this in the text.

---

## [Author Response]

Reviewer #1:[…] In summary, the technical quality of this study is very high and the authors address a very interesting question. Unfortunately, the authors have not used their RNA-binding PRC2 mutants to address the physiological role of RNA-binding for PRC2 function. Thus, it appears that the authors have stopped half-way, and publication in eLife cannot be recommended.

While we agree that in vivo studies are important, they are beyond the scope of the current study. The epigenetic silencing field has been waiting for the identification of RNA-binding sites of PRC2 for a long time, and this is a very competitive field. By publishing this analysis now, we will facilitate the in vivo studies of many groups around the world.

Reviewer #2:[…] I only have a few minor comments of editorial nature that the authors should address.1) Figure 3. This is radioactive signal and so it represents H3 molecules that were mono-, di- or tri-methylated in the reaction. The band should therefore be labelled as H3K27me and not as H3K27me3. Alternatively, the authors should provide WB data with an H3K27me3-specific antibody.

As suggested, the original label “H3K27me3” has been changed to “H3K27me”.

2) Figure 7 is not a very strong dataset. The entire figure could go into the supplement.

The original Figure 7 has been moved to the supplement (Figure 6—figure supplement 3).

3) Figure 8 is a nice summary. So it would be helpful to clearly indicate in the figure that panel B is C.t. PRC2 and C is H.s. PRC2. It may also be helpful to include a C.t and H.s. EZH2 domain architecture diagram (i.e. like in Figure 2), indicating the location of the relevant residues along the polypeptide chain.

“ctPRC2” and “hsPRC2” labels have been added in the figure, and an EZH2 domain architecture diagram has been added as Figure 7 (original Figure 8) with the location of key residues marked.

4) Discussion, first paragraph: I don't think that the term "RNA binding domain" is helpful in this case. The term "domain" should be reserved for a portion of a protein with a defined fold. Here, the authors show that the interaction of PRC2 with RNA occurs via positively charged amino acid residues that are dispersed over a large protein surface and are not even in a conserved location on the complex when comparing C.t. and H.s. PRC2.

“Domains” has been changed to “sites” to be consistent with the terms used two sentences earlier (“RNA binding to these proteins seems to occur at multiple sites of the protein.”)

5) The authors state that "missense mutations of some the H.s. EZH2 residues critical for RNA binding have been identified in cancer patients." The https://portal.gdc.cancer.gov website lists about 150 different amino acid changes that have been identified in EZH2 (which is a 746 amino acid protein) and almost all of them – including the residues mentioned in the text – were identified a single time and in completely different tumours. So it is not clear how meaningful these specific particular mutations are. Moreover, I assume that these are all heterozygous mutations in tumours where the homologous chromosome carries the EZH2 wild-type allele. Please clarify this in the text.

We totally agree with the comments that it is not clear how meaningful these specific cancer mutations are, and a few sentences have been added in the paragraph to explain the limited information for the biological consequences of these mutations.

In addition to addressing the above comments, we made several other revisions to make the story clearer and more convincing:

1) We added new results in Figure 6 and associated texts in the conclusion part (subsection “Identification of Key Residues on hsPRC2 for RNA Binding”, last paragraph). These new results complete the story by combining the identified mutations in a single mutant, and we show that RNA binding is further diminished compared to the individual point mutants.

2) To test these mutations in the context of the holoenzyme (5m hsPRC2), we added new results in Figure 6—figure supplement 2. The mutant binding to RNA is significantly weaker, but it still assembles well and doesn’t show as much defect in DNA binding.

3) We also fixed the color labeling of E1 and E2 region in Figure 5. The numbering of E1 and E2 is correct in Figure 5, but the original color labeling of these two regions was shifted in the original manuscript.

4) Materials and methods sections have been updated.

5) Supplemental figures in the original manuscript file have been incorporated with each main figure to become “Figure X—figure supplement X”.